# Hyperspectral ultraviolet to shortwave infrared characteristics of marine-harvested, washed ashore and virgin plastics

Shungudzemwoyo P. Garaba [1] and Heidi M Dierssen [2,3]

[1]Marine Sensor Systems Group, Institute for Chemistry and Biology of the Marine Environment, Carl von Ossietzky University
of Oldenburg, Schleusenstraße 1, Wilhelmshaven 26382, Germany
[2] Department of Marine Sciences, Avery Point Campus, University of Connecticut, 1080 Shennecossett Road, Groton, CT
06340, USA
[3] Institute of Material Science, Storrs Campus, University of Connecticut, 97 North Eagleville Road, Storrs, CT 06269-3136,
USA

*Correspondence to*: Shungudzemwoyo P. Garaba (shungu.garaba@uni-oldenburg.de)

## Abstract

Combating the imminent environmental problems associated with plastic litter requires a synergy of monitoring strategies, clean-up efforts, policymaking and interdisciplinary scientific research. Lately, remote sensing technologies have been evolving into a complementary monitoring approach that might have future applications in the operational detection and tracking of plastic litter at repeated intervals covering wide geo-spatial areas. We therefore present a dataset of Lambertian-equivalent spectral reflectance measurements from the ultraviolet (UV, 350 nm) to shortwave infrared (SWIR, 2500 nm) of synthetic hydrocarbons (plastics). Spectral reflectance of wet and dry marine-harvested, washed ashore and virgin plastics was measured outdoors with a hyperspectral spectroradiometer. Samples were harvested from the major accumulation zones in the Atlantic and Pacific Ocean suggesting a near representation of plastic litter in global oceans. We determined a representative bulk average spectral reflectance for the dry marine-harvested microplastics dataset available at https://doi.org/doi:10.21232/jyxq-1m66 (Garaba and Dierssen, 2019c). Similar absorption features were identified in the dry samples of washed ashore plastics, dataset available at https://doi.org/doi:10.21232/ex5j-0z25 (Garaba and Dierssen, 2019a). The virgin pellets samples consisted of eleven polymer types typically found in floating aquatic plastic litter, dataset available at https://doi.org/doi:10.21232/C27H34 (Garaba and Dierssen, 2017). Magnitude and shape features of the spectral reflectance collected were also evaluated for two scenarios involving dry and wet marine-harvested microplastics, dataset available at https://doi.org/doi:10.21232/r7gg-yv83 (Garaba and Dierssen, 2019b). Reflectance of wet marine-harvested microplastics was noted to be lower in magnitude but had similar spectral shape to the one of dry marine-harvested microplastics. Diagnostic absorption features common in the marine-harvested microplastics and washed ashore plastics were identified at ~931, 1215, 1417 and 1732 nm. In addition, we include metrics for a subset of the marine-harvested microplastic related to particle morphology including sphericity and roundness. These datasets are also expected to improve and expand the scientific evidence-based knowledge on optical characteristics of common plastics found in aquatic litter. Furthermore, these datasets

have potential use in radiative transfer simulations exploring the effects of plastics to ocean colour remote sensing and developing algorithms applicable to remote detection of floating plastic litter.

## 1 Introduction

The amount of plastic litter in the natural environment is growing exponentially, and this challenge has led to a huge demand for integrated and sustainable monitoring strategies (Lebreton et al., 2018;Maximenko et al., 2016;G20, 2017;Werner et al., 2016). Remote sensing is a widely considered tool that can provide a complementary avenue of wide geo-spatial and spectral information about plastics in natural waters (Maximenko et al., 2016). Current key requirements expected from remote sensing are to detect, identify, quantify, and track floating plastics. Feasibility studies centred on these four requirements have shown promising prospects in remote sensing of floating and submerged litter (Garaba et al., 2018;Aoyama, 2018;Kako et al., 2012;Topouzelis et al., 2019). Although current efforts are promising, there is a need to advance remote sensing of plastics and adapt future sensors to generate plastic related end-products. In line with this, a new generation of satellite missions (e.g. PRISMA – Italian Space Agency, EnMap – German Aerospace Centre, PACE – National Aeronautics and Space Administration) is anticipated to advance remote sensing of the environment through hyperspectral observations from the ultraviolet (UV, ~350 nm) to shortwave infrared (SWIR, ~2500 nm). While these future missions are not dedicated to plastic litter studies, they are likely to provide essential knowledge of high quality, hyperspectral, wide geo-spatial coverage information pertinent to plastics. Going forward, many satellite missions will be supported by observations from unmanned aerial systems, aircrafts and high altitude pseudo-satellites.

A limited number of high quality hyperspectral measurements of plastic types found in marine litter have been reported or are in open-access repositories. We, therefore, conducted measurements from the UV – SWIR with the goals to contribute towards (i) creating a high quality baseline hyperspectral reflectance dataset of weathered plastics being washed ashore or floating in the oceans (ii) identifying absorption features of naturally weathered plastics, (iii) demonstrating the high reflectivity of plastics compared to other optically active constituents of the oceans, (iv) creating an open-access spectral reference library for improved radiative transfer simulations and (v) proposing algorithms essential to 'detect, identify, quantify, track' plastics as unique from other floating debris. We present the detailed steps that were completed to acquire these measurements of the virgin types and naturally weathered plastics found in marine and land-based litter.

## 2 Methods and Materials

### 2.1 Samples

We used a set of specimens consisting of macroplastics (diameter > 5 mm) and microplastics (diameter < 5 mm). The macroplastics were collected during clean-up activities along the west coastline of the United States of America (USA), now

being used to create awareness under the theme 'Washed Ashore: Art to Save the Sea', a travelling art exhibition in USA. At the time of experiment around midday on 25 March 2015 the exhibit was at the Mystic Aquarium in Connecticut. We believe these objects (buoys, handles, bottle caps, containers, styrofoam, ropes, toys, diving fins and nets) had undergone natural weathering at sea and/or on land based on careful visual inspection with particular interest on shapes, type of original object

and colour. The macroplastics had different colour shades ranging from blue, green, yellow, orange, peach, beige, ivory to white (Garaba and Dierssen, 2018).

Marine-harvested microplastic samples were obtained from the west North Atlantic ocean using a Neuston net (mesh size = 335 µm) in the top 0.25 m seawater layer (Law et al., 2010). After collection with the nets, the microplastics were left to dry

followed by hand separation before storage in glass scintillation vials at Sea Education Association archives. Additional marine-harvested microplastic specimens used in this study were collected from Kamilo Point in Hawaii, USA by Bill Robberson and Anna-Marie Cook (Environmental Protection Agency, USA). The Kamilo Point samples were not sieved as was done for the North Atlantic samples due to the quantity that was available, we therefore classified them as aggregated microplastics. Dry virgin pellets of varying opacity were chosen to represent the polymer source types that have been identified

in specimens harvested from sediments and aquatic sampling (GESAMP, 2015;Hidalgo-Ruz et al., 2012). The polymer types considered were polyvinyl chloride (PVC), polyamide or nylon (PA 6.6 and PA 6), low-density polyethylene (LDPE), polyethylene terephthalate (PET), polypropylene (PP), polystyrene (PS), fluorinated ethylene propylene teflon (FEP), terpolymer lustran 752 (ABS), Merlon, polymethyl methacrylate (PMMA).

The variability in apparent colour and shape of the marine-harvested microplastics and washed ashore macroplastics is a plausible representation of the plastic litter that is being found in the aquatic and terrestrial environment but may not necessarily represent all the plastic litter found globally.

## 2.2 Spectral reflectance measurements

Hemispherical directional reflectance measurements (Nicodemus et al., 1977) of all specimens were conducted outdoors during
daylight hours ± 3 hours around midday using an Analytical Spectral Devices (ASD) FieldSpec® 4 hyperspectral spectroradiometer (Malvern Panalytical, USA) between 350 and 2500 nm. An 8.5° foreoptic was used during the spectral measurements of the dry macroplastics at a 45° nadir angle 10 cm above target object. A 99 % Spectralon® Lambertian plaque (Labsphere, USA) was used for white referencing and optimizing the integration time of the spectroradiometer. It was assumed that by using a Spectralon® Lambertian plaque for white referencing we eliminated the effects of varying setup geometry

during measurements. Spectra were recorded first over the plaque then from 5 different spots of each dry macroplastic object and then over the plaque. A single spectral measurement was derived as an average of 20 continuous automated scans. Microplastics were aggregated on a black rubber mat to create an optically dense target before each spectral measurement (**Figure 1**). This black rubber mat was used as background target because it had a negligible spectral reflectance over the

spectrum range observed. At a 0° nadir angle, a 1° foreoptic was placed 8 cm above the aggregated microplastics on the black mat and reference measurements were made using a Spectralon® Lambertian plaque (Labsphere, USA). Again here, a spectrum was first collected over the plaque followed by 10 measurements over the aggregated dry microplastics and then finally over the plaque. Before each measurement over the dry microplastics, we gently mixed the particles to rearrange the orientation and location of the particles in an effort to get a best representative bulk spectral signal. A similar approach was used to perform additional spectral measurements of wet microplastics in filtered seawater with a salinity of 30 ppt. Our selected setup was determined to be optimum and minimized instrument and user shading.

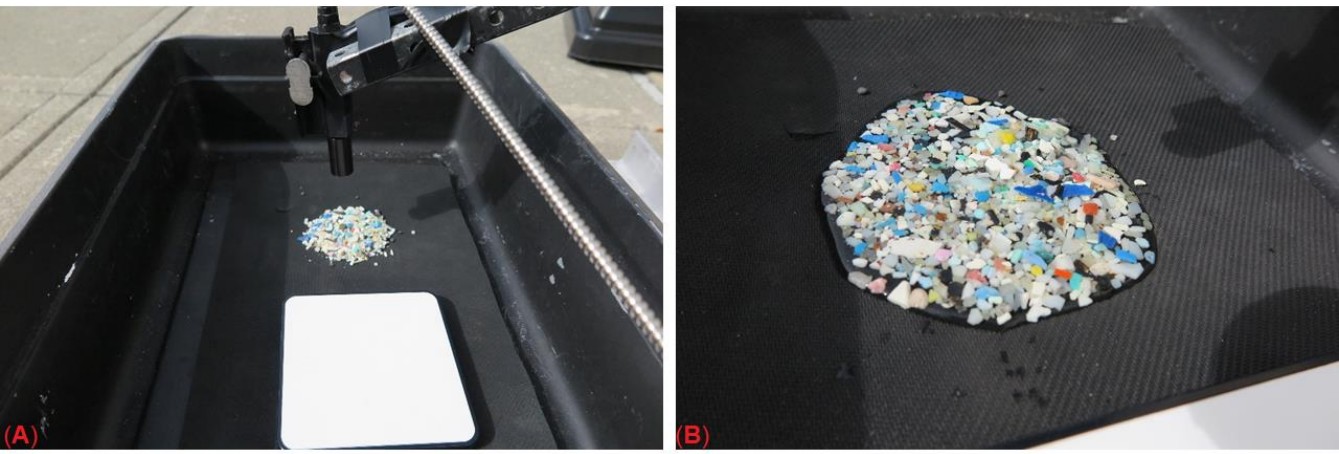

**Figure 1** Experimental setup with the aggregated (A) dry and (B) wet marine harvested microplastics. A black neoprene rubber was used as a background in a dark spray painted container to mitigate background light during spectral reflectance measurements.

## 2.5 Data processing

Lambertian-equivalent spectral reflectance ($R$) was calculated as the ratio of the measurement taken over the sample divided by that taken over the plaque normalized by the spectral calibration of the plaque. Natural outdoor lighting allowed us to measure spectral reflectance with good signal-to-noise from the UV to SWIR, with the exception of certain SWIR regions where the atmosphere is opaque. These regions of the spectrum are shown as gaps in the continuous spectrum, from 1350 nm to 1410 nm and 1800 nm to 1950 nm in the dataset. Average spectra were calculated for each set of repeated measurements. All data processing, statistics and plots were generated in MathWorks MATLAB.

### 2.5.1 Spectral absorption features

In general, the spectral reflectance of an optically active object (e.g. plastic, corals, seawater) has a characteristic shape that explains how it can reflect or absorb light. The spectral shape is a combination of peak (reflection or fluorescence) features and trough (absorption) features that are distinctive optical properties of the objects. An absorption feature would occur at spectral wavebands where the object absorbs more light and reflects less light compared to the neighbouring wavebands. Here,

*a priori* knowledge about typical absorption features in hydrocarbons or plastics was combined with visual inspection of measured $R$ (**Figure 2a**). Further verification of these identified absorption features was done by applying a moving average filter with a window of 19 nm to derive a smoothed $R$. Second order derivatives of the smoothed spectra were then computed to generate derivative $R$ signals with enhanced absorption features (**Figure 2b**). Using derivatives of spectra has been shown to be a robust analytical tool in remote sensing (Dierssen et al., 2015;Huguenin and Jones, 1986;Russell et al., 2016;Tsai and Philpot, 1998). Continuum removal was applied to the $R$ followed by calculating a relative band-depth index to enumerate the absorption intensity (**Equation 1**). An end and start waveband was obtained from a MathWorks MATLAB R2016a convhull function. The function objectively locates the convex hull i.e. wavebands immediately before ($\lambda_1$) and after ($\lambda_3$) the absorption feature waveband (**Figure 2c**). The equation used to calculate the band-depth (bd) at the central wavelength ($\lambda_2$) from the reflectance at the three wavebands is:

$$bd = R_1 - R_2 + \left(\frac{\lambda_2 - \lambda_1}{\lambda_3 - \lambda_1}\right) * (R_3 - R_1) \tag{1}$$

Band-depth indexes are widely used as proxies for detection and quantification of target optically active objects in natural environments (Clark, 1983;Clark, 1999;Dierssen, 2019). Absorption features are thus enhanced after being normalized by the continuum removal approach (**Figure 2d**). Diagnostic absorption features refer to those parts of the spectrum unique to a particular object such that they possess a similar shape and are located at a specific wavelength range (Clark et al., 2003). An inter-comparison to check for diagnostic absorption features was conducted using the macroplastic and microplastic spectra.

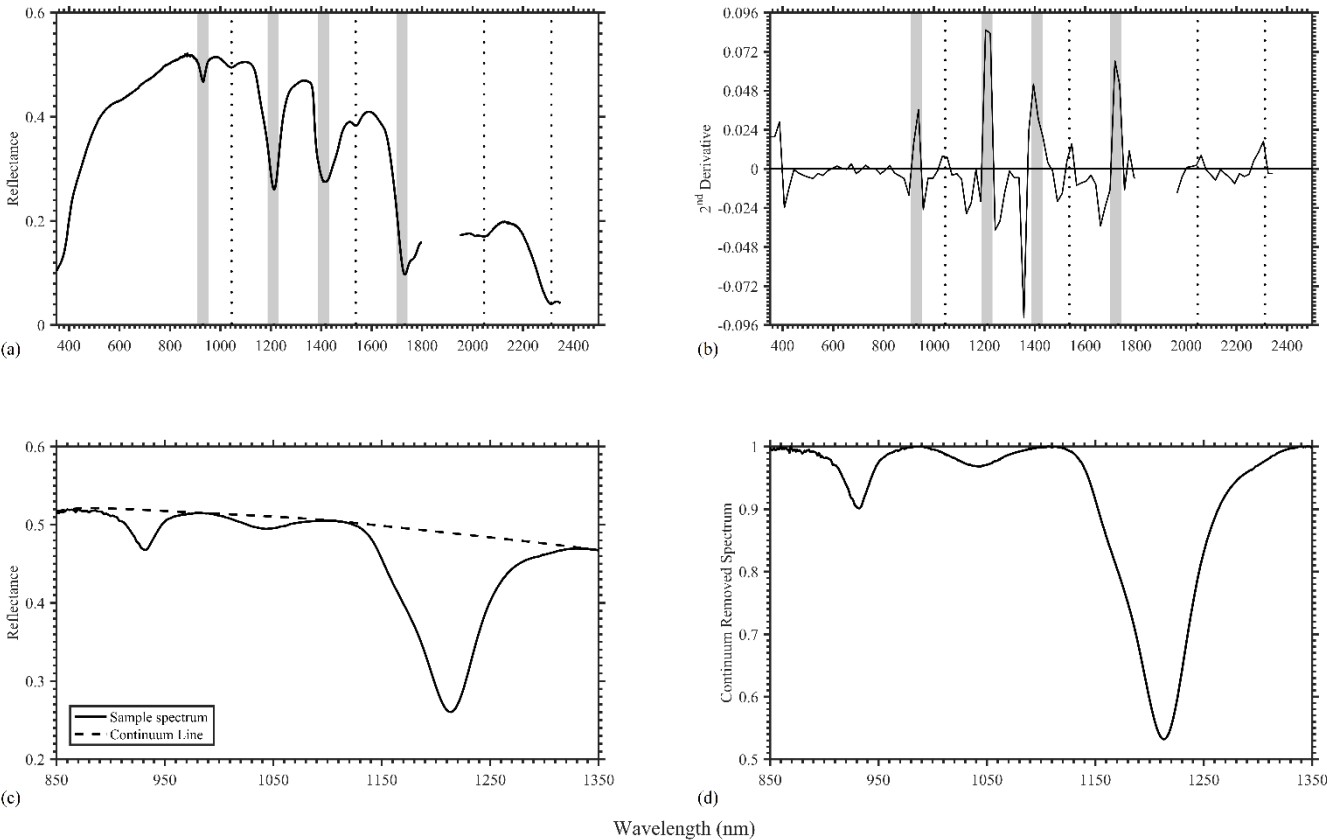

**Figure 2.** (a) Example spectral reflectance used for visual inspection to identify absorption features highlighted by the vertical lines, (b) second derivative signal validating the location of absorption features, (c) continuum line generated from the convhull function and (d) continuum removed signal.

5 The degree of spectral shape likeness among the measured $R$ was calculated using a quantitative similarity scoring algorithm (Wan et al., 2002). A spectral contrast angle ranging from (0° = very strong degree of similarity to 90° = no similarity) was determined after converting the spectra of two samples into a multi-dimensional vector that is not affected by the inherent intensity of the spectra but only depends on the shape (**Equation 2**). Assuming x and y to be reflectance at a given wavelength of a sampled and reference target the spectral contrast angle (Θ) is derived as,

$$\Theta = \cos^{-1} \frac{\sum x \cdot y}{\sqrt{\sum x^2 \sum y^2}} \tag{2}$$

A scale to evaluate spectral shape similarity classified the results of **Equation 2** as very strong (0° $\leq \Theta \leq$ 5°), strong (5° < Θ $\leq$10°), moderate (10° < Θ$\leq$ 15°), weak (15° < Θ$\leq$ 20°) and very weak (20° <Θ) (Garaba and Dierssen, 2018).

**2.5 Microplastic particle morphology**

A Marathon electronic digital calliper was used to measure the size distribution of dry marine-harvested microplastic particles. Additional particle descriptors included sphericity, roundness, and a qualitative description of colour. Particle sphericity and roundness were determined according to a qualitative scale (Powers, 1953).

5 **3 Results**

**3.1 Macroplastics**

Spectral reflectance of the different dry washed ashore macroplastics (Garaba and Dierssen, 2019a) had significant differences in the visible spectrum related to the intrinsic colour of each object (**Figure 3**). Blue objects peaked around 450 nm, green objects around 550 nm while white objects had a flatter reflectance in visible wavelengths. Beige and ivory coloured object

10 had rapidly increasing reflectance in the visible with an eightfold reflectance magnitude rise from 400 nm to 700 nm. Yellow, peach and orange object also had increasing reflectance in the visible but not as pronounced as in the beige and ivory objects, ranging from a three to fourfold increase in reflectance. Overall, the highest reflectance was noted on the beige object $R = 0.88$ around 850 nm. For all the objects, the reflectance peaked in the NIR followed by a general decrease in the SWIR with several absorption features resulting in localized dips and peaks. Despite the variations in the spectral magnitude and shape, absorption

15 features common to all the macroplastics were located at wavebands centred close to 931 nm, 1045 nm, 1215 nm, 1417 nm, 1537 nm, 1732 nm, 2046 nm and 2313 nm (**Figure 3**). The location of the absorption features was validated and confirmed by derivative analyses of each respective spectrum.

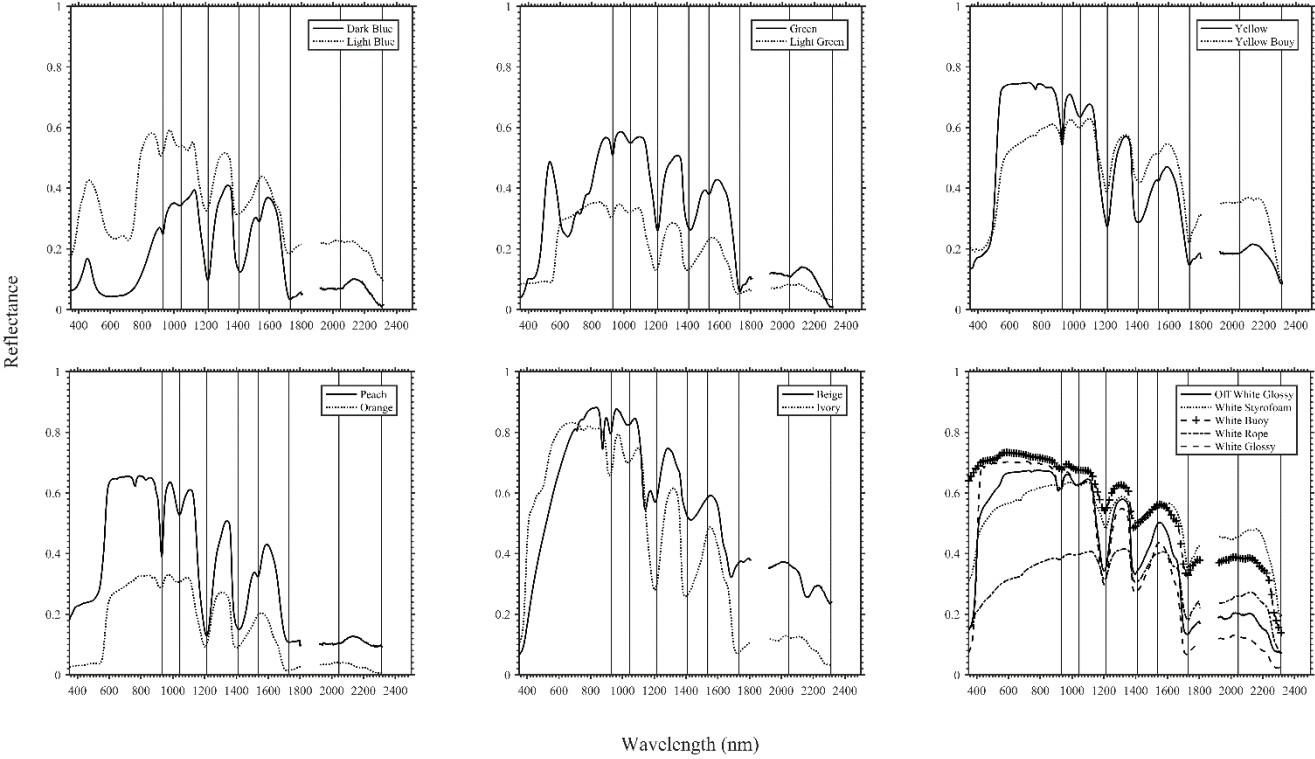

**Figure 3.** Spectral reflectance of dry washed ashore macroplastics harvested along the western coast of the USA. Absorption features noted in marine-harvested microplastics are highlighted by the vertical lines (centred around ~931 nm, 1045 nm, 1215 nm, 1417 nm, 1537 nm, 1732 nm, 2046 nm, 2313 nm).

### 3.2 Marine-harvested microplastics

Spectral reflectance of the dry marine-harvested microplastics (Garaba and Dierssen, 2019c) increased with wavelength reaching highest values in the NIR at 850 nm then decrease towards the SWIR wavebands. All spectra were close to uniform in both spectral shape (mean $\Theta < 5°$) and magnitude (percentage ranges $< 40$ %) compared to the macroplastics. A non-parametric Kruskal-Wallis one-way analysis of variance was utilized to determine if any differences existed in the measured spectra of the dry marine-harvested microplastics. The statistical test suggested no significant differences ($p < 0.05$) in the spectral reflectance from 350 to 2500 nm. We therefore determined a representative dry marine-harvested microplastic spectral endmember (**Figure 4**). Absorption features identified in the dry washed ashore macroplastic specimens (**Figure 3**) were also found in the dry marine-harvested microplastics (**Figure 4**).

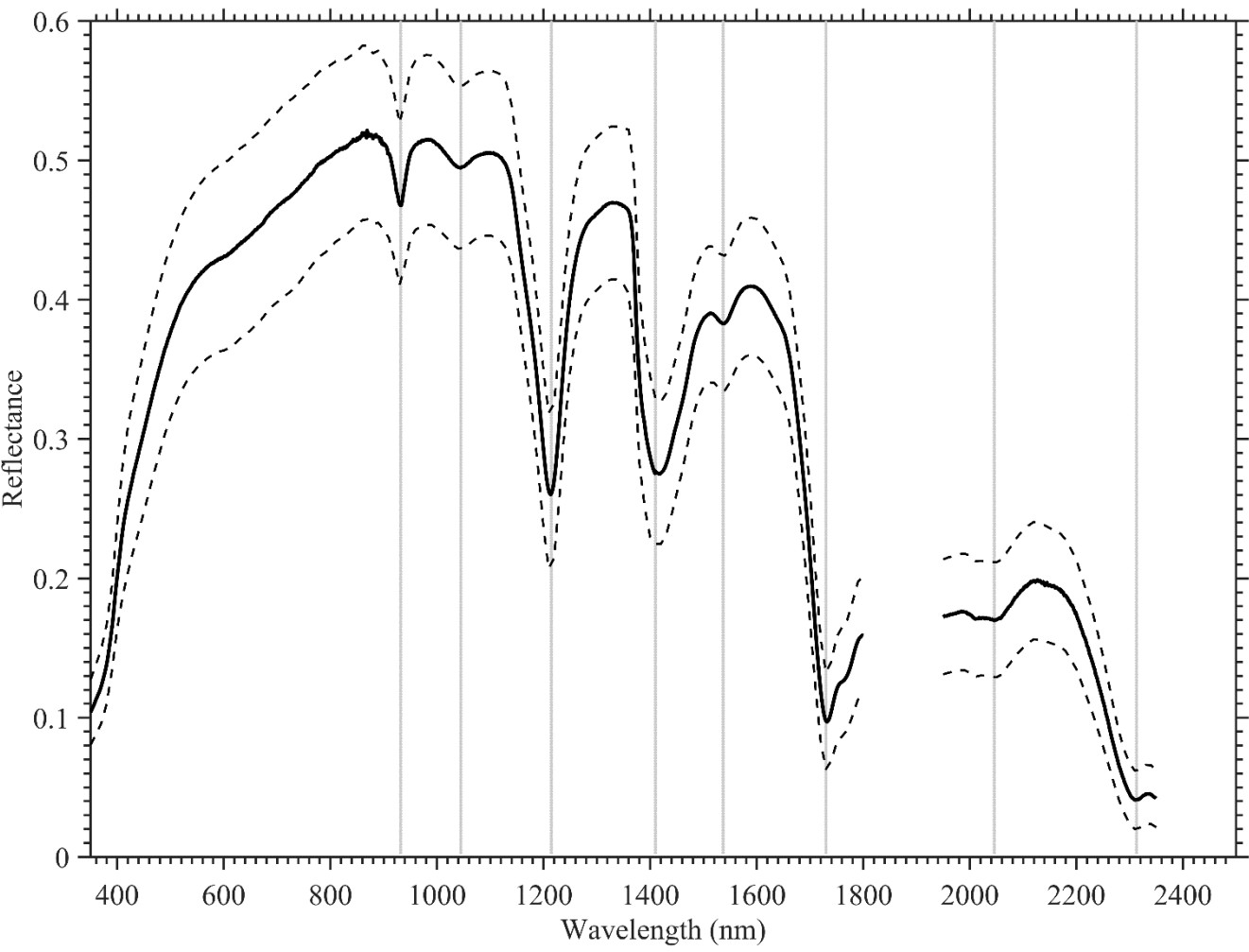

**Figure 4.** Endmember spectral reflectance with 1 standard deviation continuous error bars of the dry marine-harvested microplastics. Identified unique absorption features are highlighted by the vertical lines provide the wavebands that are outlined in grey (centred around ~931 nm, 1045 nm, 1215 nm, 1417 nm, 1537 nm, 1732 nm, 2046 nm, 2313 nm).

Wet marine-harvested microplastics (Garaba and Dierssen, 2019b) had similar absorption features found in the dry marine-harvested specimens (**Figure 5**). However, the magnitude of $R$ decreased by 12 % in the UV to 90 % in the SWIR due to the presence of water mixed with the samples. The loss of reflectance in the plastics was observed to be consistent with the increase in the absorption coefficient of pure water (**Figure 5b**). Average decrease in the measured R was 56 ± 23 %. In addition, the spectral absorption features were less pronounced in the wet samples compared to the dry and some were not noticeable (1045, 1537, and all >2000)

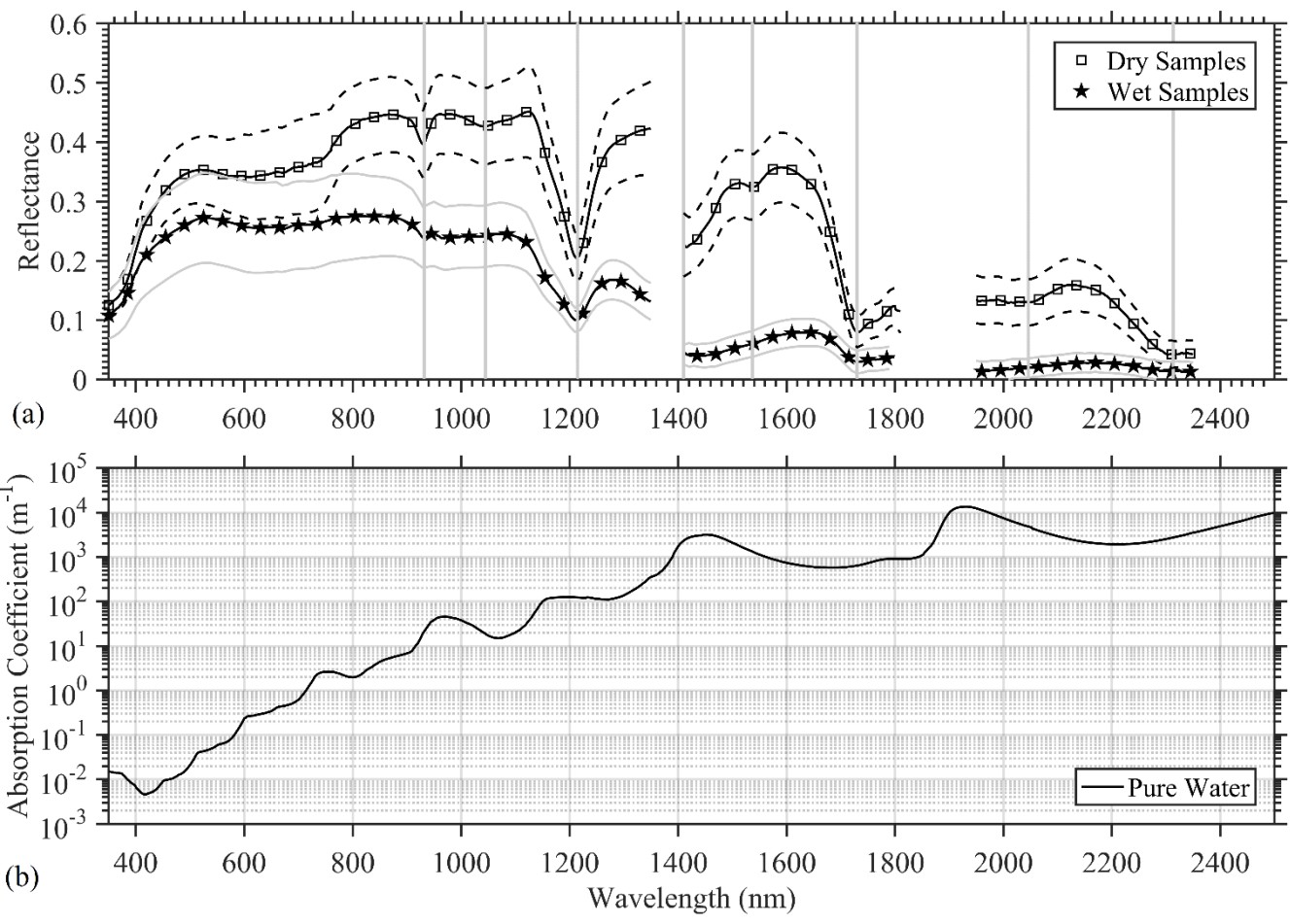

**Figure 5.** (a) Mean bulk spectral reflectance of dry and wet marine-harvested microplastics with 1 standard deviation continuous dashed error bars. Absorption features noted in marine-harvested microplastics are highlighted by the vertical lines (centred around ~931 nm, 1045 nm, 1215 nm, 1417 nm, 1537 nm, 1732 nm, 2046 nm, 2313 nm). (b) Absorption coefficient of pure water (Rottgers et al., 2011).

### 3.3 Virgin pellets

Spectral properties of the dry virgin pellets (Garaba and Dierssen, 2017) varied in magnitude and shape (**Figure 6**). However, two specimens of PA (6 and 6.6) did show very strong similarities ($\Theta = 2.1°$), although the apparent opacity of PA 6 was lower than that of PA 6.6. FEP was generally flat in the NIR to SWIR. Overall, the highest reflectance was observed in the specimens

10 of ABS, PMMA and PVC while the lowest was observed in PET. Only FEP and PVC were noted to have a strong reflectance in the SWIR > 1900 nm with $R > 0.4$. Several of the absorption features from the marine-harvested and washed ashore specimens were duplicated in the dry virgin pellets, although some features were absent e.g. FEP or shifted compared to the marine plastics (**Figure 6**). We also determined that our marine-harvested microplastic endmember was best matched to PMMA, PP, LDPE and PET (Garaba and Dierssen, 2018).

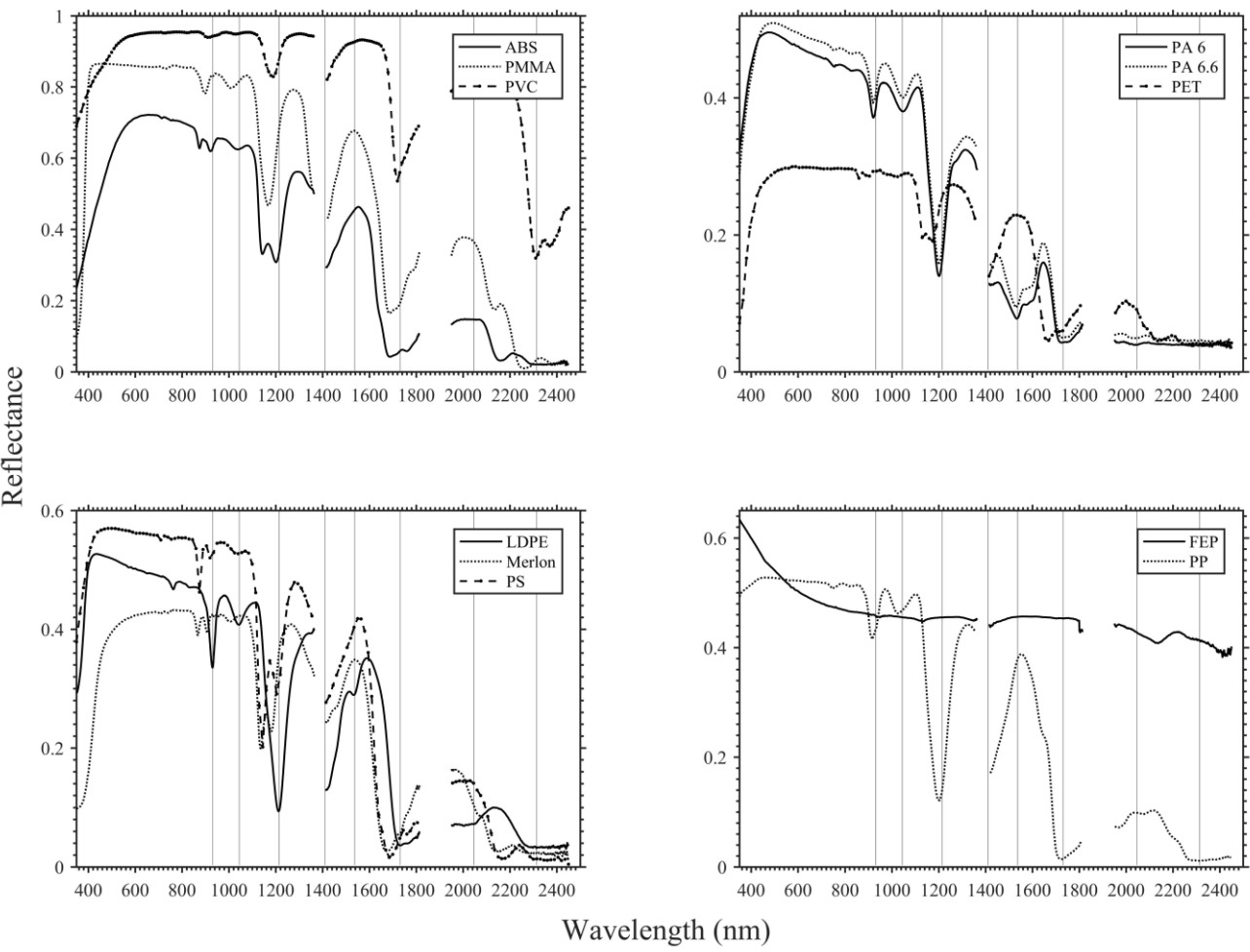

**Figure 6.** Spectral reflectance of dry virgin pellets and absorption features found in marine-harvested and washed ashore plastics highlighted by the vertical lines. Absorption features identified in marine-harvested microplastics are highlighted by the vertical lines (centred around ~931 nm, 1045 nm, 1215 nm, 1417 nm, 1537 nm, 1732 nm, 2046 nm, 2313 nm).

## 3.4 Morphology of dry marine-harvested microplastics

Morphometric measurements were completed on a total of 47 microplastic particles from different size classes (**Table 1**). The particles were brittle and could fracture with handling. Sphericity of the observed particles ranged from low to high sphericity whilst the roundness was between sub-angular to very angular (**Table 1**). Dry virgin pellets common ocean plastic litter had varying opacity of the colour white. **Table 1** is available as an excel sheet as supplementary material.

**Table 1.** Microplastic particle side distribution and colour.

| Length (mm) | Width (mm) | Height (mm) | Colour | Sphericity | Roundness |
| --- | --- | --- | --- | --- | --- |
| 7.72 | 3.25 | 2.44 | green | high | very angular |
| 9.96 | 5.59 | 1.94 | white | low | very angular |
| 9.34 | 3.81 | 2.36 | white | low | very angular |
| 3.94 | 4.56 | 1.54 | white | low | very angular |
| 7.17 | 4.57 | 1.50 | green | high | very angular |
| 5.65 | 3.76 | 2.16 | white | low | very angular |
| 3.97 | 3.43 | 3.14 | white | low | sub-angular |
| 7.56 | 4.48 | 0.90 | white | low | very angular |
| 6.71 | 4.11 | 1.91 | white | high | very angular |
| 7.45 | 4.15 | 2.22 | green | high | very angular |
| 6.50 | 4.32 | 1.90 | white | low | very angular |
| 8.08 | 4.51 | 2.04 | white | low | very angular |
| 6.86 | 4.03 | 1.56 | white | low | very angular |
| 6.64 | 3.63 | 0.82 | green | low | very angular |
| 7.55 | 4.62 | 0.71 | white | low | very angular |
| 5.51 | 4.91 | 1.59 | white | low | very angular |
| 8.50 | 4.46 | 1.61 | black | low | very angular |
| 5.63 | 4.49 | 1.90 | white | low | very angular |
| 8.32 | 4.45 | 2.00 | white | low | very angular |
| 6.28 | 4.04 | 1.03 | white | low | very angular |
| 4.26 | 3.94 | 1.77 | white | high | sub-angular |
| 5.55 | 4.94 | 1.40 | white | high | very angular |
| 5.04 | 3.79 | 0.90 | white | high | very angular |
| 5.89 | 4.43 | 0.91 | white | high | very angular |
| 6.47 | 4.57 | 1.75 | white | low | very angular |
| 13.07 | 3.82 | 0.42 | white | low | very angular |
| 4.50 | 4.29 | 1.35 | white | low | sub angular |
| 4.62 | 4.35 | 2.09 | white | low | sub angular |
| 6.57 | 4.69 | 1.37 | white | low | very angular |
| 17.21 | 4.58 | 0.36 | white | low | very angular |
| 7.77 | 5.32 | 0.60 | white | low | very angular |
| 5.19 | 4.28 | 1.26 | white | high | very angular |
| 3.85 | 2.71 | 2.62 | white | high | very angular |
| 6.14 | 4.41 | 0.85 | white | high | very angular |
| 10.54 | 3.66 | 0.35 | white | low | very angular |
| 5.59 | 4.25 | 1.62 | white | high | very angular |
| 8.29 | 4.66 | 1.01 | white | low | very angular |
| 10.16 | 4.15 | 0.89 | white | low | very angular |

| | | | | | |
|---|---|---|---|---|---|
| 15.08 | 4.38 | 0.25 | white | low | very angular |
| 7.33 | 4.34 | 0.68 | white | high | very angular |
| 6.30 | 4.00 | 0.61 | white | high | very angular |
| 8.52 | 5.77 | 0.42 | white | high | very angular |
| 7.17 | 5.64 | 1.23 | white | high | very angular |
| 4.93 | 4.28 | 1.09 | white | high | sub-angular |
| 6.35 | 5.34 | 0.77 | white | low | very angular |
| 6.68 | 3.67 | 0.62 | white | low | very angular |
| 7.64 | 4.46 | 0.65 | white | high | very angular |

## 4 Discussion

We measured Lambertian-equivalent spectral reflectances of washed ashore, marine-harvested as well as virgin plastics and identified 8 absorption features (centred around ~931 nm, 1045 nm, 1215 nm, 1417 nm, 1537 nm, 1732 nm, 2046 nm, 2313 nm) in most of the weathered specimen. Location of these absorption features agreed well with prior reports (Asadzadeh and

de Souza Filho, 2017;Hörig et al., 2001). Of these 8 wavebands, we concluded that ~931 nm, 1215 nm, 1417 nm, 1732 nm were diagnostic absorption features after continuum removal and derivative analyses. Several studies have already shown prospective applications of the ~1215 nm and ~1732 nm wavebands in detection and quantification algorithms for floating and land-based plastic litter (Garaba et al., 2018;Goddijn-Murphy and Dufaur, 2018;Kühn et al., 2004). Unfortunately, the 931 nm and the 1417 nm absorption features fall outside the atmospheric window. These features pose a challenge for algorithm

development as the plastic specific signal will be scrambled in the signal from atmospheric gases especially water vapour around ~900 nm and ~1400 nm. We also simulated the potential detection of submerged plastics and observed a decrease in the measured $R$ which was consistent with the enhanced absorption of light by pure water in the SWIR (**Figure 5**). One aspect that was not addressed with the dataset was the depth at which the submerged plastic can be detected, it is important to further study the optical properties of plastic litter with varying water depth. Due to enhanced water absorption, spectral features in

the NIR and SWIR region will quickly disappear as particles submerge and only reflectance in visible wavelengths would be detectable with remote sensing.

The polymer characterization of the specimen is a key factor in advancing scientific evidence-based knowledge complemented by spectral measurements as it enables researchers to further create essential descriptors for remote sensing applications related

to plastic litter. Laboratory techniques are typically used to accurately determine polymer compositions of marine-harvested or washed ashore plastics, these include Fourier Transform Infrared (FTIR), Raman spectroscopy and pyrolysis gas chromatography (Thevenon et al., 2014). However, in our case the washed ashore macroplastics were part of an ongoing travelling plastic litter awareness exhibit and no detailed analyses could be conducted to determine polymer composition. The only descriptors obtained from the washed ashore macroplastics were the object color, shape/form and spectral reflectance.

Future campaigns are recommended to collect additional high quality descriptors (polymer composition, refractive index, date

of manufacture, sphericity, individual size distribution) of plastic specimen that will improve classification and radiative transfer modelling efforts. It is also important to consider the possibilities to expand the spectral reference library through spectral unmixing simulations to create blended polymers from the virgin pellets.

Variability in the spectra was reported within one standard deviation and a median was also determine to be consistent with literature (Dierssen, 2019;Russell et al., 2016;Zibordi et al., 2011). However, future measurements should include comprehensive uncertainty budgets to enable advanced error propagation efforts when data is assimilated into radiative transfer models. Implementing the algorithms that use spectral shape and continuum removal approaches reduces the uncertainties related to variations in magnitude. These investigations should also explore the possible anisotropic optical properties of plastic
litter especially as it breaks down and weathers in the natural environment.

**4 Data availability**

Quality control was performed according to the guidelines of SeaDataNet. All the datasets are in open-access via the online repository EcoSIS spectral library https://ecosis.org/. The dry marine-harvested microplastics spectral data are available at https://doi.org/doi:10.21232/jyxq-1m66 (Garaba and Dierssen, 2019c). Washed ashore plastics spectral data is available at
https://doi.org/doi:10.21232/ex5j-0z25 (Garaba and Dierssen, 2019a). The virgin pellets spectral data is available at https://doi.org/doi:10.21232/C27H34 (Garaba and Dierssen, 2017). Dry and wet marine-harvested microplastics spectral dataset available at https://doi.org/doi:10.21232/r7gg-yv83 (Garaba and Dierssen, 2019b).

**5 Conclusions and outlook**

We have established an open-access dataset of hyperspectral reflectances of dry washed ashore macroplastics, dry marine-harvested microplastics, artificially wetted marine-harvested microplastics and virgin pellets. The dataset provides some of the first baseline measurements that can be assimilated into radiative transfer modelling to improve scientific knowledge on the contribution of plastic litter to the bulk signal reaching remote sensing sensors. Furthermore, such knowledge about the hyperspectral characteristics of micro and macroplastics litter can be used to evaluate the capabilities and application of current
multispectral sensors in remote sensing efforts. Using spectral response functions of current remote sensing tools (airborne, unmanned aerial systems, high altitude pseudo-satellites, satellites) it is also possible to simulate the spectral signature of our dataset, this information would be crucial in the design of future or planned remote sensing tools.

**Author contribution**

SPG and HMD contributed equally to the experiment and manuscript preparation.

## Competing interests

The authors declare that they have no conflict of interest.

## Acknowledgements

Funding was provided by NASA Ocean Biology and Biogeochemistry through the PACE project (NNX15AC32G) and Deutsche Forschungsgemeinschaft (DFG, German Research Foundation) – Projektnummer 417276871.

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
