# Peer review of "Hyperspectral ultraviolet to shortwave infrared characteristics of marine-harvested, washed ashore and virgin plastics"

_Earth System Science Data, 2019_

## Referee Comment (RC1) · Elizabeth C Atwood (Referee) · 1 Oct 2019

**General Comments**

The submitted data description paper contains important baseline measurements of both macro- and microplastics, the latter in both weathered, dry and wet, as well as virgin form. This is an important dataset for the field of remote sensing of marine plastics and the authors are highly applauded for making such a dataset public. We all thank you for making this available to everyone. I still have some comments as to how

this data should be presented, especially with respect to questions that remain open as to what these data represent.

**Specific Comments**

Light has different penetration depths dependent on wavelength. This has been somewhat accounted for in the analysis in that wet microplastics were also measured, but a direct discussion of this fact is missing from the entire manuscript (I made note of this being missing at lines 20/21 in the Abstract and at the end of Section 3.2). This point should be addressed in more depth in the manuscript.

It would also help to make the above-mentioned point if in Fig. 4 the 30 ppt saltwater absorption curve over the same wavelengths range would be shown. Furthermore, the maximum penetration depth of light at each particular wavelength could be presented. This would aid in better representing the limitations of the dataset.

There is no discussion of what different plastic types are represented in the microplastic samples and in what proportions. I noted this first at line 10 in Section 2.1, where the samples are being separated by size (which never comes up again in the following analyses, so that step is somewhat moot? Consider removing it). This is especially frustrating since you thereafter present the spectral curves for different virgin plastic types. The discussion of microplastic spectral curves (first paragraph Section 3.1) focuses only on different colors of the plastic measured, despite the fact that different polymer types play a very important role for spectral curve shape in the NIR-SWIR range. The plastic types of the macroplastics needs to be listed.

I furthermore find it misleading to use 1 standard deviation in Fig. 3 and 4 to represent the variability in the data. The 1 s.d. curve is only truly representative of the data distribution if the data are normally distributed. In my experience, spectral measurements at any particular wavelength more often tend to be skewed in one direction or the other. Given this issue, I find the upper and lower percentage errors much more informative (one could still use the +/- 34.1% quantile around the mean to remain close to the 1

s.d. idea).

In Fig. 4, it is confusing that the error curves for the different sampling curves overlap and look exactly the same (both exactly the same type of dashed line).

Section 3.3 brings back the importance of my point above about proportion of different polymer types in the microplastic samples. Given the measured virgin polymer plastic curves, one could perform a spectral unmixing analysis of the dry and wet microplastic samples to determine proportion of different plastic types. One could then validate this separately by analyzing the microplastic samples down to plastic type. Maybe it doesn't make a difference to know what the different proportions of polymer types are in a sample, but I haven't come across a convincing study yet that makes this point.

**Technical Corrections**

No technical corrections were found needed.

---

## Referee Comment (RC2) · Anonymous Referee #2 · 17 Oct 2019

This paper presents reflectance spectra for various types of plastics in the marine environment (dry vs wet, macro and micro, washed ashore and marine collected) plus virgin plastic pellets. While spectral reflectance at an arbitrary viewing geometry and illumination environment is not a sufficient input for formal radiative transfer simulations as the authors suggest, these spectra do have value in that they can be used to identify spectral absorption features of potential value for remote sensing.

We follow the review guidelines for the ESSD journal, indicated in italics.

[Figure]

*Read the manuscript: Are the data and methods presented new? Is there any potential of the data being useful in the future? Are methods and materials described in sufficient detail? Are any references/citations to other data sets or articles missing or inappropriate?*

Overall, the data and methods are new, useful, and presented in sufficient detail. A few terms could use a bit more explanation to make this paper useful as a stand alone product. For example: 1. the Spectral Shape Similarity is described in Garaba and Dierssen 2018, but I think this would be much more powerful if equation 1 from that paper was also included here. 2. I feel that the sphericity and roundness scale from Powers 1953 is outdated and inappropriate for use if the goal is radiative transfer simulation a simple aspect ratio would suffice. Table 1 has this, great, but was that table in the data files? Perhaps I missed it. 3. Derivative analysis needs to be defined. Again, this is in the Garaba and Dierssen 2018 paper, but definition is needed here. I find the description of what exactly constitutes a spectral 'feature' lacking (both here and the 2018 paper), and the listing of these features inconsistent in this paper. For example, the abstract lists four, apparently strong features. Section 3.1 shows eight. Qualitatively, I also question a few of the features – 2046 seems inconsistent, and 2313 seems too close to the edge of the spectral range to be valid. 1417 seems too close to other (water?) absorption features to be useful.

*Is the article itself appropriate to support the publication of a data set?*

Yes, with the modifications noted above.

*Check the data quality: Is the data set accessible via the given identifier? Is the data set complete? Are error estimates and sources of errors given (and discussed in the article)? Are the accuracy, calibration, processing, etc. state of the art? Are common standards used for comparison?*

The data do appear available as noted, although the link for the "dry washed ashore macroplastics" appears broken in the abstract (but not the link in section 4). As far as I

can tell, no uncertainty metrics were provided for the ASD observations, a key missing component. If this exists, it should be included. Although the paper notes differences in viewing geometry, foreoptic aperture and spectralon plaque reflectance for both the macro and micro plastics, no explanation for these different choices was given. This should be rectified.

*Is the data set significant – unique, useful, and complete?*

Yes

*Consider article and data set: Are there any inconsistencies within these, implausible assertions or data, or noticeable problems which would suggest the data are erroneous (or worse). If possible, apply tests (e.g. statistics). Unusual formats or other circumstances which impede such tests in your discipline may raise suspicion.*

The data set, to the best of my ability to confirm, looks good.

*Is the data set itself of high quality?*

yes

*Check the presentation quality: Is the data set usable in its current format and size? Are the formal metadata appropriate? Check the publication: Is the length of the article appropriate? Is the overall structure of the article well structured and clear? Is the language consistent and precise? Are mathematical formulae, symbols, abbreviations, and units correctly defined and used? Are figures and tables correct and of high quality?*

Yes to all of these

*Is the data set publication, as submitted, of high quality?*

yes

*Finally: By reading the article and downloading the data set, would you be able to*

*understand and (re-)use the data set in the future?*

Yes, if the above issues are resolved. This could be used to help identify spectral absorption features for qualitative remote sensing algorithms, but not for input to radiative transfer simulations as the authors suggest. The latter requires spectrally resolved complex refractive indicies, measurements of size distribution and sphericity. Hopefully these data will be measured in the future.

---

## Author Comment (AC1) · 19 Nov 2019

**Reply to reviewer**

| | |
|---|---|
| **Manuscript Title** | : Hyperspectral ultraviolet to shortwave infrared characteristics of marine- |
| | : harvested washed ashore and virgin plastics |
| **Authors** | : Garaba S.P. and Dierssen H. M. |
| **Journal** | : Earth System Science Data *(ESSD)* |

**Elizabeth C Atwood (Referee)**
**Referee comment - 1**
**General Comments**
The submitted data description paper contains important baseline measurements of both macro- and microplastics, the latter in both weathered, dry and wet, as well as virgin form. This is an important dataset for the field of remote sensing of marine plastics and the authors are highly applauded for making such a dataset public. We all thank you for making this available to everyone.
**Author response - 1**
We appreciate your kind words and taking the time to carefully review our manuscript.

**Referee comment - 2**
I still have some comments as to how this data should be presented, especially with respect to questions that remain open as to what these data represent.
**Specific Comments**
Light has different penetration depths dependent on wavelength. This has been somewhat accounted for in the analysis in that wet microplastics were also measured, but a direct discussion of this fact is missing from the entire manuscript (I made note of this being missing at lines 20/21 in the Abstract and at the end of Section 3.2). This point should be addressed in more depth in the manuscript.
**Author response - 2**
Thank you for pointing this out. We have appended text to elucidate more on this (See section *4 Discussion* of the revised manuscript).

**Referee comment - 3**
It would also help to make the above-mentioned point if in Fig. 4 the 30 ppt saltwater absorption curve over the same wavelengths range would be shown. Furthermore, the maximum penetration depth of light at each particular wavelength could be presented. This would aid in better representing the limitations of the dataset.
**Referee comment - 3**
We have now included the absorption coefficient of pure water to further explain the wavebands affected by absorption of water (See *Figure 5* and *4 Discussion* of the revised manuscript).

**Referee comment - 4**
There is no discussion of what different plastic types are represented in the microplastic samples and in what proportions. I noted this first at line 10 in Section 2.1, where the samples are being separated by size (which never comes up again in the following analyses, so that step is somewhat moot? Consider removing it). This is especially frustrating since you thereafter present the spectral curves for different virgin plastic types.
**Author response - 4**
The text has been removed as suggested.

**Referee comment - 5**
The discussion of microplastic spectral curves (first paragraph Section 3.1) focuses only on different colors of the plastic measured, despite the fact that different polymer types play a very important role for spectral curve shape in the NIR-SWIR range. The plastic types of the macroplastics needs to be listed.

**Author response - 5**
That is correct, polymer type plays a key factor in the spectral shape. Future sampling is expected to consider recording these additional critical descriptors: object color, shape, polymer composition, date of manufacture, dimensions. We now further acknowledge this limitation in the revised manuscript (See section *4 Discussion* of the revised manuscript).

**Referee comment - 6**
I furthermore find it misleading to use 1 standard deviation in Fig. 3 and 4 to represent the variability in the data. The 1 s.d. curve is only truly representative of the data distribution if the data are normally distributed. In my experience, spectral measurements at any particular wavelength more often tend to be skewed in one direction or the other. Given this issue, I find the upper and lower percentage errors much more informative (one could still use the +/- 34.1% quantile around the mean to remain close to the 1 s.d. idea).

**Author response - 6**
We agree that it can be misleading to provide datasets with 1 standard deviation. We added text to make this caveat clear in the manuscript (See section *4 Discussion* of the revised manuscript).

**Referee comment - 7**
In Fig. 4, it is confusing that the error curves for the different sampling curves overlap and look exactly the same (both exactly the same type of dashed line).

**Author response - 7**
We improved the representation of the figure (See *Figure 5* of the revised manuscript).

**Referee comment - 8**
Section 3.3 brings back the importance of my point above about proportion of different polymer types in the microplastic samples. Given the measured virgin polymer plastic curves, one could perform a spectral unmixing analysis of the dry and wet microplastic samples to determine proportion of different plastic types. One could then validate this separately by analyzing the microplastic samples down to plastic type. Maybe it doesn't make a difference to know what the different proportions of polymer types are in a sample, but I haven't come across a convincing study yet that makes this point.

**Author response - 8**
A good point, the spectra from these virgin pellets can be used to expand the spectral reference library by performing unmixing with various combinations and proportions of the polymers. It is an area of interest to be further investigated (See section *4 Discussion* of the revised manuscript).

**Technical Corrections**
No technical corrections were found needed.

---

## Author Comment (AC2) · 19 Nov 2019

**Reply to reviewer**

| | |
|---|---|
| **Manuscript Title** | : Hyperspectral ultraviolet to shortwave infrared characteristics of marine- |
| | : harvested washed ashore and virgin plastics |
| **Authors** | : Garaba S.P. and Dierssen H. M. |
| **Journal** | : Earth System Science Data *(ESSD)* |

**Anonymous Referee #2**

**Referee comment - 1**
This paper presents reflectance spectra for various types of plastics in the marine environment
(dry vs wet, macro and micro, washed ashore and marine collected) plus virgin plastic pellets. While spectral reflectance at an arbitrary viewing geometry and illumination environment is not a sufficient input for formal radiative transfer simulations as the authors suggest, these spectra do have value in that they can be used to identify spectral absorption features of potential value for remote sensing.
**Author response - 1**
Thank you for taking the time to review and provide constructive feedback on our manuscript. We agree that investigations on the anisotropic distribution of the spectral reflectance of marine-harvested plastics is needed for more accurate simulations and hope future works will consider this (*See section 4 Discussion of the revised manuscript*).

**We follow the review guidelines for the ESSD journal, indicated in italics.**
**Read the manuscript: Are the data and methods presented new? Is there any potential of the data being useful in the future? Are methods and materials described in sufficient detail? Are any references/citations to other data sets or articles missing or inappropriate?**
**Referee comment - 2**
Overall, the data and methods are new, useful, and presented in sufficient detail. A few terms could use a bit more explanation to make this paper useful as a stand alone product. For example:
1. the Spectral Shape Similarity is described in Garaba and Dierssen 2018, but I think this would be much more powerful if equation 1 from that paper was also included here.
**Author response -2**
Thank you for pointing this out. We have appended the methods section by explaining the data processing steps used and appended Equation 1 on spectral shape similarity (*See section 2.4 Data processing of the revised manuscript*).

**Referee comment -3**
2. I feel that the sphericity and roundness scale from Powers 1953 is outdated and inappropriate for use if the goal is radiative transfer simulation a simple aspect ratio would suffice.
Table 1 has this, great, but was that table in the data files? Perhaps I missed it.
**Author response - 3**
Yes, Table 1 was provided in the manuscript and as supplementary material

**Referee comment - 4**
3. Derivative analysis needs to be defined. Again, this is in the Garaba and Dierssen 2018 paper, but definition is needed here. I find the description of what exactly constitutes a spectral 'feature' lacking (both here and the 2018 paper), and the listing of these features inconsistent in this paper. For example, the abstract lists four, apparently strong features. Section 3.1 shows eight. Qualitatively, I also question a few of the features – 2046 seems inconsistent, and 2313 seems too close to the edge of the spectral range to be valid. 1417 seems too close to other (water?) absorption features to be useful.
**Author response - 4**

-We have included information about derivative analysis and provide a definition of a spectral feature and expand on the methodology used to objectively identify the spectral features reported (*See section 2.5.1 Spectra absorption feature of the revised manuscript*).

-A new figure to summarize the spectral data analyses has been included (*See Figure 2 of the revised manuscript*).

-We have revised the text to clarify the mismatch in the number of absorption features in the abstract and in the results. In the abstract we have added the term 'diagnostic' which means an absorption feature is unique to a particular material in shape and is located around a limited wavelength range (*See Line 27 of the revised manuscript abstract, sections 2.5.1 Spectra absorption feature and 4 Discussion of the revised manuscript*).

-Yes, the 1417 nm feature is affected by atmospheric gases. The other features (centred around 2046, 2313 nm) have also been report in other studies. We have added text to point out the useful wavebands and studies that have already utilized several of the wavebands reported here (*See section 4 Discussion of the revised manuscript*).

**Is the article itself appropriate to support the publication of a data set?**
**Referee comment - 5**
Yes, with the modifications noted above.
**Author response -5**
Thank you and we have carefully made revisions to improve clarity of the text.

**Check the data quality: Is the data set accessible via the given identifier? Is the data set complete? Are error estimates and sources of errors given (and discussed in the article)? Are the accuracy, calibration, processing, etc. state of the art? Are common standards used for comparison?**
**Referee comment 6**
The data do appear available as noted, although the link for the "dry washed ashore macroplastics" appears broken in the abstract (but not the link in section 4).
**Author response -6**
The links have been checked and updated.

**Referee comment - 7**
As far as I can tell, no uncertainty metrics were provided for the ASD observations, a key missing component. If this exists, it should be included.
**Author response - 7**
We agree a comprehensive uncertainty budget is missing and is needed. However, the datasets are provided with the descriptive statistics mean, median and standard deviation. Each measurement was an average of 20 scans.

**Referee comment - 8**
Although the paper notes differences in viewing geometry, foreoptic aperture and spectralon plaque reflectance for both the macro and micro plastics, no explanation for these different choices was given. This should be rectified.
**Author response - 8**
We assume by doing a white reference with the Spectralon plaque after white referencing we obtained a Lambertian Equivalent Reflectance suggesting minimal effect by geometry. We have added text to clarify this (*See section 2.2 Spectral reflectance measurements of the revised manuscript*).

**Is the data set significant – unique, useful, and complete?**
**Referee comment - 9**

Yes

**Consider article and data set: Are there any inconsistencies within these, implausible assertions or data, or noticeable problems which would suggest the data are erroneous (or worse). If possible, apply tests (e.g. statistics). Unusual formats or other circumstances which impede such tests in your discipline may raise suspicion.**

**Referee comment - 10**

The data set, to the best of my ability to confirm, looks good.

**Is the data set itself of high quality?**

**Referee comment - 11**

Yes

**Check the presentation quality: Is the data set usable in its current format and size? Are the formal metadata appropriate? Check the publication: Is the length of the article appropriate? Is the overall structure of the article well structured and clear? Is the language consistent and precise? Are mathematical formulae, symbols, abbreviations, and units correctly defined and used? Are figures and tables correct and of high quality?**

**Referee comment - 12**

Yes to all of these

**Is the data set publication, as submitted, of high quality?**

**Referee comment - 13**

Yes

**Finally: By reading the article and downloading the data set, would you be able to understand and (re-)use the data set in the future?**

**Referee comment -14**

Yes, if the above issues are resolved. This could be used to help identify spectral absorption features for qualitative remote sensing algorithms, but not for input to radiative transfer simulations as the authors suggest. The latter requires spectrally resolved complex refractive indicies, measurements of size distribution and sphericity. Hopefully these data will be measured in the future.

**Author response -14**

Thank you and yes further measurements are warranted to support detailed radiative transfer models and suggest future works should consider such investigations *See section 4 Discussion of the revised manuscript*).